# Influence of Mineral Fertilizer and Manure Application on the Yield and Quality of Maize in Relation to Intercropping in the Southeast Republic of Kazakhstan

**DOI:** 10.3390/plants11192644

**Published:** 2022-10-08

**Authors:** Maksat Batyrbek, Fakher Abbas, Ruqin Fan, Qingfang Han

**Affiliations:** 1College of Agronomy, Northwest A & F University, Xianyang 712100, China; 2College of Resources and Environment, Zhongkai University of Agriculture and Engineering, Guangzhou 510225, China

**Keywords:** maize, mineral fertilizer, poultry manure, crop yield, intercropping

## Abstract

Maize (*Zea mays* L.) is a valuable forage crop. It is also an essential and promising crop for the Republic of Kazakhstan, cultivated in the southern zone. Some new maize hybrids have been introduced, which have been beneficial for high yields with less fertilizer input. This study aims to introduce the new maize hybrid, Arman 689, for the judicial use of fertilizer and the high yield. This study was carried out in 2015 in the southeast region of Kazakhstan. There are five treatments with various mineral fertilizer and poultry manure doses: 1. control (T0), 2. P_60_ K_100_ (T1), 3. N_100_P_60_K_100_ (T2). 4. N_100_P_60_K_100_ + 40 tons of manure/ha (T3), and 5. N_100_P_60_K_100_ + 60 ton of manure/ha (T4). The fertilizers used were ammonium nitrate (N—34.6%), amorphous (N—11.0%, P_2_O_5_—46.0%), and potassium chloride KCl (K_2_O—56%). The results showed that the grain yield ranges from 5.51 t/ha (T0) to 8.49 (T4) t/ha. The protein contents in the maize grain varied from 9%(T0)–11.3%(T4). The grain nitrogen content accounted for 54.2 to 52.0%. The nutrient uptake results by different treatments indicated that nitrogen contributed to 41.5% of the total yield increase. Using manure in combination with mineral fertilizers reduced the payback of the applied resources, as the payback of T2–T4 was 8.8–9.1 kg of grain. With the application of recommended mineral fertilizer (NPK), the protein yield was 0.83 t/ha, 0.33, and 1.22 t/ha higher than T0 and T1 treatments, respectively. There was no significant yield difference under T3 and T4 treatments (*p* > 0.05). Overall, the treatment, NPK + 40 tons of manure, was proved the ultimate for the Arman hybrid in providing the optimum quantity and quality of maize, as well as reducing the payback cost (8.8–9.1 kg of grain). It is suggested to apply NPK-recommended doses along with manure in maize (Arman hybrid)-based intercropping systems to utilize the resources efficiently.

## 1. Introduction

Maize (*Zea mays* L.) is one of the valuable forage crops. It contributes to at least 30 percent of the food demand of around 4.5 billion people in developing countries [1]. In developing countries, the development of fodder crops is critical for livestock production. Compared to other crops, maize performed exceptionally well in the cold season; therefore, fodder maize production in the autumn overcomes the problem of animal feed shortages [2]. By its versatility, maize surpasses almost all forage crops. Both green mass and grains are used as fodder. It is nutritional fodder that is liked by most animals, especially those used for the purpose of milk production [3]. The nutritional requirements of crops are fulfilled with the use of scientific-based doses of mineral or/and organic fertilizers. The adequate availability of nutrients, especially primary nutrients: nitrogen (N), phosphorus (P), and potassium (K), are essential components of the cropping system, and their availability at different growth stages of the crop is vital [4]. Providing optimal and recommended doses of fertilizers is directly linked with a stable yield increase of the crops. Improving plant nutrition helps mobilize the plant’s physiological resources and increase the yield. However, each crop has a specific limit to applying the nutrients. Maize belongs to a group of exhaustive crops, which means that after each harvest, it consumes nitrogen (N), phosphorus (P), and potassium (K) [5]. With a yield of 6–7 tons of grains or 50–70 tons of green mass per hectare, it takes out about 150–180 kg of N, 50–60 kg of P_2_O_5_, and 150–200 kg of K_2_O from the soil. The potential requirement for nutrients is a hereditary trait of maize hybrids [6]. A considerable variation (21 to 58%) for maize was recorded using phosphorus (P) by mineral or organic fertilizer [7]. The wide variability in K and N content was also recorded along with P. Many authors indicated that hybrids and varieties of different crops do not respond similarly: late-ripening types are more responsive to fertilizers than early maturing types [8,9,10,11]. In order to sustain the economic stability of a country and the survival of the human race, it is critical to manage food security. According to a projection, there will be an additional population of approximately 9 billion people by the year 2050, which would result in a greater need for food, particularly legumes and cereals. Nevertheless, increasing industrial development restricted the growth of grain production, particularly in emerging nations, where there is limited agricultural land with a greater population. Intercropping is one of the ways to resolve these food security issues due to higher resource efficiency: more output with less input [12,13]. Intercropping is a centuries-old multiple cropping technique that is widespread amongst small-scale farmers in the developing world [14,15]. Among all the combinations of intercropping, growing legumes (soybean, chickpea, peanut) in between the maize is the most suitable option [16] due to its greater adaptation to varying agroecological factors and varying climatic conditions [17,18]. Maize is also the most suitable option for intercropping, as it offers broad space for the proper growth of any other crop, especially legumes [19]. Maize is an essential and promising crop for the Republic of Kazakhstan, mainly cultivated in the southern zone of the Republic of Kazakhstan. The grain yield average ranges from 5.24 to 6.14 t/ha. Scientists have given some scientific recommendations on using fertilizers for grain purposes of maize cultivation [20,21]. The recommendations were based on the peculiarities of the crop, the type of soil on which the maize is cultivated, nutritional elements, and the planned yield level. This work aimed to study the effect of mineral fertilizer and manure application on the (1) chemical composition of plants and the nutrient uptake in the early plan development, and (2) on the grain quality of the mid-season maize hybrid, Arman 689.

## 2. Materials and Methods

### 2.1. Location of the Study

This study was conducted on light chestnut irrigated soils in the southeast of the Republic of Kazakhstan, in the zone of the foothill plain of the Zailiyskiy Alatau. The location of the experiment was the Kazakh Scientific Research Institute of Agriculture and Plant Growing (KSRIAP). In terms of texture, the soil was coarse silty to medium loam. The foothill steppe zone was located at an altitude of 650–700 m above sea level. The average annual precipitation in this area is 54.84 mm, with fluctuations of 6.1 to 112.7 mm. The annual average air temperature recorded is +52.65 °F (worldwheater). The sand, silt, and clay contents of the soil ranged between 40–45%, 8.6%, and 43–45%, respectively (Table 1). Overall, micro-aggregates varied between 80–90% for different treatments. The organic matter (OM) content was not adequate for soils, 1.9–2.6%. The soil carbonate content was high enough; the pH of soils was slightly alkaline, 7.5–7.8. The absorption capacity (AC) varied between 15.0–16.0 mg^−eq^. The total nitrogen (TN) content of the soils was about 0.15%, total phosphorus (TP) was 0.21%, alkaline-hydrolyzable nitrogen was 140 mg/kg, mobile phosphorus was 20–20.5 mg/kg, and exchangeable potassium content ranged between 400–450 mg/kg soil.

### 2.2. Experimental Design and Description

The experiment was started in April 2015, and harvesting was performed in August 2015. The local-approved hybrids of the selection, Budan (Arman 689), were chosen from the Maize Institute “Zimun Pope” (Republic of Serbia). Arman 689 is a simple interline mid-season hybrid that belongs to the FAO 600 ripening group, which takes 125–130 days from germination to full ripeness of the grain This study was a section of a five-year project (2015–2019) to evaluate the effect of mineral and organic fertilizers on maize hybrid (Arman 689) cultivation on light chestnut soil. As the hybrid, Arman 689, was recently introduced in this region, and since the findings of the 1st year of any study are of great significance, we hereby present the results of year 2015 only. Considering the role of intercropping for resolving the global food supply issue, from 2016 to 2019, along with former treatments (2015), maize-bean intercropping was also adopted for a better evaluation of this maize hybrid (Arman 689) affected by fertilizers (mineral and organic) and pair crops. The results of 2016–2019 will be published separately. The design of the experiment was a randomized complete block design (RCBD) with four replications for each treatment. The plot size for each treatment was 14 m* 15 m. The field faced a prolonged application of mineral fertilizers. Sowing was carried out at the recommended time when the soil was warmed up to 10–12 °C (in the 3rd decade of April). The sowing pattern was one-line, with a 70 cm distance between the rows. The seeding rate was 66–70 thousand/ha. The fertilizers used were ammonium nitrate (N 34.6%), amorphous (N 11.0%, P_2_O_5_ 46.0%), and potassium chloride KCL (K_2_O—56%). The experimental scheme included: 1. control (T0), 2. P_60_K_100_ (T1), 3. N_100_ P_60_ K_100_ (T2), 4. N_100_ P_60_ K_100_ + 40-ton manure (T3), 5. N_100_, P_60_, K_100_ + 60-ton manure (T4) application. Nitrogen fertilizers were applied before sowing as flooding (N_60_) and in the form of top dressing (N_40_) after the emergence of 5–6 leaves. Phosphorus, potash, and manure were applied at the bed preparation stages with the above-mentioned recommended rates.

### 2.3. Basic Characteristics of the Soil Used

Before the ploughing and sowing operations, random soil samples were collected across the field for a better evaluation of the experimental site. Soil samples were collected with augur up to the depths of 0–20 and 20–40 cm across different locations in the field. Later, the samples from two depths were made as composite samples for analysis. Debris was removed, and samples were sieved through a <2 mm sieve. The primary nutrients in the soil were determined before the experiment according to the relevant state standards. The geographical characteristics of the field experiments corresponded to those accepted for the given zone, and crops differed in the high genetic potential of productivity. Soil pH was determined by a pH meter (Mettler Toledo) using a soil-to-water ratio of 1:15 [22]. Soil organic carbon (Orc) was estimated by the oxidation method of potassium di-chromate method [23], and total N (Tn) was estimated by a micro-Kjeldahl apparatus [24]. The water content (Wc) of soils was calculated by taking the fresh weight and drying it in the oven and taking the dry weight, and then the final weight was calculated based on the initial fresh weight and dry weight [25]. The percentage of primary soil particles (clay, silt, and sand) was determined by the particle size analyzer meter [26]. Mobile phosphorus was calculated according to the Michigan method in modification of the Central Research Institute of Agrochemical Services for Agriculture State Standard 26205-91. Exchangeable K was calculated on a flame photometer following State Standard 26205-91. The content of nitrogen, potassium, and phosphorus was determined after wet-ashing of maize samples, with the subsequent determination of nitrogen by the Kjeldahl method [27], colorimetric phosphorus [28], and potassium on a flame photometer [29]. The microbial biomass of nitrogen (MBN) and carbon (MBC) were calculated by a fumigation–extraction method, where some soil samples were fumigated and stored in the extractor, and others were kept as the control with any treatments. An extract was later taken with filter paper, and the MBC and MBN were calculated [30].

### 2.4. Maize Yield

To collect the yield and growth data from the field, the recommended procedure was adopted. From each plot, 50 random plants were selected, their plant height was calculated, and their average was measured. The number of total cobs of corn was divided by the number of plants in each replicated plot. Out of all cobs/plots, 50 cobs were selected, their number of rows per cob and number of grains per row were calculated, and the mean value was estimated. From every plot, 50 samples of 1000 grains were randomly selected, and their weight was calculated. Finally, after complete shelling, the total grain yield of each plot was calculated and the average of four repeats was averaged. The yield was converted to t/ha. The dry biomass of the crop was determined by the gravimetric method [31]. At the complete maturity of the maize crop (after 147 days), the crop was harvested from each pot. The mathematical processing of the yield data was carried out using Dospekhov’s method [32].

### 2.5. Statistical Analysis

Based on the one-way analysis of variance (ANOVA), all the results of this study were evaluated. For a thorough evaluation of differences between the treatments, the statistical software, Statistics 8.1 (Analytical, Tallahassee, FL, USA), was used. This study’s design was a randomized complete block design (RCBD). The comparison of the treatments was analyzed using the least significant difference (LSD) test with a *p*-value at <0.05. The registered software, Origin Pro 8, was used for designing tables and graphs.

## 3. Results

### 3.1. Pre-Sowing Soil Characteristics

The results of this study explored that the input of complete mineral fertilizers (NPK) led to an increase in the content of alkaline-hydrolyses nitrogen in the 0–20 cm layer of soil by 19 mg kg^−1^. The combined application of mineral and organic fertilizers (60 t/ha) increased the alkaline-hydrolyses nitrogen content by 32 mg/kg. In contrast, in the soil with no fertilizer application, it was limited to 141 mg kg^−1^. The content of mobile phosphorus with complete mineral fertilization increased by up to 22.6 mg kg^−1^, whereas the combined use of organic and mineral fertilizers increased by 48.9 mg kg^−1^ of soil. The content of exchangeable potassium with mineral fertilizers increased by 64 mg kg^−1^ of soil. The combined application of organic and mineral fertilizers increased to 84 mg kg^−1^, whereas it was 464 mg kg^−1^ in the control. The microbial biomass carbon (MBC) and microbial biomass nitrogen (MBN) content of treatments ranged between 2.1 to 9.4 and 40.2 to 180.2 mg kg^−1,^ respectively. The highest and lowest values of MBC and MBN were observed for NPK + 60 t manure application and control treatment (*p* < 0.05), respectively (Table 2).

### 3.2. The Effect of Mineral and Organic Fertilizers on the Biomass of Maize

The results showed that the gross weight of the g/50 maize plants in the control treatment during the initial growth stage (formation of 5–6 leaves) was 41.8 g (Table 3). Overall, 96 g/50 plants of biomass were recorded under NPK treatment, with 82 g/50 plants against PK treatment, which showed a positive effect of N on the yield of the studied maize. The biomass of the NPK + 60 t manure was significantly higher (*p* < 0.05) than all the other treatments, whereas the lowest was observed under the control treatment. With complete mineral fertilizers (NPK) and poultry manure (40 and 60 tons) application, dry biomass was increased by 2.3 to 2.4 times compared with the control, respectively.

### 3.3. Influence of Mineral and Organic Fertilizers on Nutrients Uptake and Consumption

According to the results in the phase of 5–7 leaves, N, P, and K uptake in the control treatment was 1.40 g/50 plants, 0.21 g/50 plants, and 1.71 g/50 plants, respectively (Table 4). The use of mineral fertilizers helped to improve plant nutrition, and assimilated nutrients. In the variants with complete mineral fertilization and the additional application of organic fertilizers, the maize plants received 3.87–4.50 g of N, 0.61–0.86 g of P, and 4.23–5.15 g of K per 50 plants. In the phase of full ripeness, the most significant accumulation of N was due to grain, and its maximum was found in the variants, NPK, NPK + 40, and NPK + 60 tons of manure. Overall, for grain, the N content varied from 54.2–52.0%. There was a high correlation coefficient (τ) between the yield (τ = 0.99) and nitrogen consumption in g/50 plants at the phase of 6–7 leaves. The correlation coefficient (τ) between the protein content and nitrogen uptake in the biomass of maize in the early development period was noted with a coefficient of τ = 0.98. The nitrogen content of the plants varied from 4.0 to 4.03% for all fertilizer treatments, whereas its content in the control (CK) was up to 3.34% (Table 3). Hence, the increase of nitrogen in other treatments than control was 19.8–20.7%. With mineral fertilizer application, the amount of P in maize tissues was increased by 1.28–1.52 times. The K content in maize plants was 4.41–4.56% for fertilized options, and 4.10% for the option without fertilizer application. In the phase of the full ripeness of maize plants, significant amounts of nitrogen and phosphorus were observed in the grain. It was estimated that plant height, shoot dry weight, and chlorophyll content of the NPK treatment supported with chlorophyll content was the highest among all five treatments.

### 3.4. Influence of Mineral and Organic Fertilizers on Grain Yield of Maize

The results of this study suggested that the highest grain (*p* < 0.05) yield was highest under N_100_P_60_K_100_ + 40 and 60 tons of manure (Table 5); there was no significant difference in grain yield among the mentioned two treatments (*p* > 0.05). The increase in grain yield in these variants relative to the control without fertilizers was 2.96–2.98 t/ha. When using P_60_K_100_, the grain yield increased by 1.46 t/ha. When applying N_100,_ the variant of P_60_K_100_, the grain yield increased compared to the control treatment by 2.29 t/ha. Compared with PK treatment, additional nitrogen (N_100_) along with the P_60_K_100_ (NPK treatment) increases the yield by 0.99 t/ha. The results showed that nitrogen accounted for 41.5% and 59.7% of the total yield increase for N_100_P_60_K_100_ PK treatment. As per the calculations, by using P_60_K_100_ and N_100_P_60_K_100_ variants, the payback of fertilizer costs was 8.8–9.1 kg of grain. It was found that the protein content in the maize grain varied from 9.0–11.3%. Applying a complete mineral fertilizer, NPK (N_100_, P_60_, K_100_), increased the protein content in the grain by 1.6%. The additional application of organic fertilizers contributed to an increase in the protein in the maize grain by 2.3% (Table 5). The protein content of the crop in the control variant was 0.5 t/ha, whereas for P_60_K_100_, it increased up to 0.68 t/ha. With the application of blended fertilizer (NPK), the protein content was 0.83 t/ha, which was higher than the control (0.33 t/ha).

## 4. Discussion

### 4.1. The Effect of Nutrient Sources on the Biomass and Yield of Maize

The grain yield of all the treatments was significantly higher than T0 (*p* < 0.05), which might be due to the combined effect of NPK and manure application. Applying mineral fertilizers along with manure significantly increased the grain yield of maize. The highest grain collection was noted when applying N_100_P_60_K_100_ + 40 (T3) and 60 (T4) tons of manure, and it was significantly higher (about 60% higher than control) than T0 (*p* < 0.05). These results follow the findings of previous studies, which indicated a positive impact of NPK along with organic fertilizer application on crop growth and yield [33,34,35,36]. Compared with the control, there was an apparent effect of the NPK fertilizer along with manure on the biomass of the maize crop in the early growth stage, as the treatment, NPK, along with 60-ton manure, showed the highest (*p* > 0.05) biomass (112.5 g), which was 2–3 times higher than the crop with the control treatment. Along with the yield, the results of this study revealed the vital role of mineral and organic fertilizer application on the maize biomass (Table 2), as the biomass under all the other treatments was significantly higher than the control (T0) (*p* < 0.05); these results were consistent with the literature [37]. The yield was highest (8.47 and 8.49 ton/ha) under T3 and T4 with manure at 40 and 60 t/ha application, respectively. These results of the high yield of maize with manure application are consistent with many studies, but are contradicted by most of the studies that applied manure from 2 to 20 tons/ha, which were believed to be sufficient to provide a high yield [37,38,39,40,41] There was no significant difference in grain yield between T3 and T4 (*p* > 0.05), indicating that NPK with manure application at 40 tons/ha and 60 tons/ha manure are similar. The T3 was more beneficial because of the low cost and was equally suitable as T4. These results are similar to the findings of Filho et al. [42], which suggested using 4 tons of manure instead of 6 and 8 tons/ha for optimal yield and maximum return. The high biomass content with NPK and manure application was consistent with previous studies [43]. Because of the growing population and resulting food demand, we need to utilize the resources (nutrients, water) smartly. Maize is considered among the cereal crops that are best suited for an intercropping-based system with leguminous crops, and is ideal for utilizing environmental resources [44]. Maize was also recommended as a better option for intercropping than the pure stand because it resulted in a higher yield and better resource utilization, and is useful against diseases and in pest management. It also provides shade to the component crop (especially legumes), and has N availability because of the fixation by legumes, soil structural stability by holding a large area, and a low risk of investment loss because of the two crops [45]. In the past, many researchers used maize as an idea paired with the leguminous crop for the judicial use of environmental resources. As such, based on the high yield using the recommended NPK dose along with 40–60 tons of manure application, we suggest using these recommended nutrients for maize–legume intercropping with the recommended spacing method. Some studies noticed intercropping (e.g., maize with legumes) showed a higher yield/net benefit than sowing as a pure stand [46,47]. The positive effect of N on the yield was also recorded. The results indicated that N alone accounted for 41.5% of the total yield increase; the positive effect of nitrogen on the yield was in line with Blumenthal et al. [48], which suggested that nitrogen boosts the biomass and yield of crops. Aside from biomass and yield, protein content is also one of the significant components of the maize crop. In this study, we observed a significant role of a full dose of NPK fertilizer with manure application on protein content, as there was an increase in protein content for T4 and T5 of 1.6 and 2.3%, respectively. The high protein content increase followed by NPK and manure application is parallel with the findings of Cai et al. [30], who described that these might be due to the high chlorophyll content. Overall, 40 t/ha manure (T3) application combined with blended fertilizer (NPK) proved to be the most suitable for improving the biomass, yield, and protein content, but also for reducing the payback cost (8.8–9.1 kg of grain).

### 4.2. Influence of Nutrient Sources on the Nutrient Use Efficiency

Our research has established that the amount of N, P, and K uptake, and the consumption, depend on the growth stages of the crop (Table 3), and nutrient content decreases with the growth stages [49]. Mineral and organic fertilizers helped improve plant growth and nutrient uptake [50]. In the current study, we also found similar results, as maize plants received 3.87–4.50 g/50 plants of nitrogen, 0.61–0.86 g/50 plants of phosphorus, and 4.23–5.15 g/50 plants of potassium, which is significantly higher than the control (T0) (Table 6). After complete ripeness, the share of nitrogen was highest for grain, from 52.0 to 54.2%. The grain yield of the plot with NPK treatment was significantly higher (*p* < 0.05) than PK treatment, which revealed a vital role of N on the grain yield of maize; similar results were found in a previous study [51,52]. A strong correlation (r = 0.98) was found between the protein content and nitrogen accumulation in the biomass of maize in the early development period, which suggested that nitrogen was the critical segment of protein production [53]. In all the treatments, NPK + 60 tons of manure was found to be more productive in all the growth stages. The use of mineral and organic fertilizers along with nitrogen content also increased the amount of phosphorus in maize tissues by 1.28–1.52 times, whereas potassium content was increased by 0.46% compared with the control treatment; this was also reported previously [54]. The microbial biomass carbon (MBC) and microbial biomass nitrogen (MBN) content of T4 treatment were the highest, which might be due to the high nitrogen content, as NPK, along with organic manure, improves soil microbial community [55]. Our results also suggested that NPK + 60 t treatment showed a significant positive effect on plant height, shoot weight, and chlorophyll content (Table 7). This was in line with the literature [56,57], which indicated the positive impact of nitrogen and manure on plant vegetative growth and chlorophyll content.

## 5. Conclusions

Applying mineral fertilizers along with manure significantly increased the grain yield of maize. The highest grain collection was noted when applying N_100_P_60_K_100_ + 40 (T3) and 60 (T4) tons of manure, and it was significantly higher (about 60%) than T0 (*p* < 0.05). Along with the yield, the results revealed the meaningful role of mineral and organic fertilizer application on the maize biomass (Table 2), as the biomass under all the other treatments was significantly higher than the control (T0) (*p* < 0.05). A significant role of nitrogen was found in all the treatments; after complete ripeness, the share of nitrogen, 52.0 to 54.2%, was highest in grain. In the total increase in yield, nitrogen accounted for 40.3%, whereas phosphorus (P) and potassium (K) fertilizers accounted for 59.7%. A high linear correlative relationship was observed between nitrogen consumption (in the 5–7 leaf phase) and grain yield and its protein content quality (τ = 0.94, 0.99). We observed a significant role of a full dose of NPK fertilizer with 40 and 60 tons of manure application on protein content; those were increased by 1.6 and 2.3%, respectively. Overall, the treatment, NPK + 40 t manure, proved to be the ultimate in providing the optimum for a high biomass, high protein content, high nutrient uptake, and, ultimately, high yield of maize, good for the quantity and quality of maize, as well as reducing the payback cost (8.8–9.1 kg of grain). It is judicial to apply NPK-recommended doses along with manure (40 t/ha) in maize (Arman hybrid)-based intercropping systems to efficiently utilize the resources with double dipping.

## Figures and Tables

**Table 1 plants-11-02644-t001:** Basic characteristic of the soil prior to sowing operation.

Parameter (Unit)	Content
Sand (%)	43.4 ± 2.3
Silt (g kg^−1^)	44.4 ± 1.4
Clay (%)	10.2± 4.5
Microaggregate (%)	85.2 ± 5.32
pH	7.7 ± 0.2
Total organic carbon (g kg^−1^)	13 ± 0.5
Total carbon (g kg^−1^)	23.4 ± 1.5
Total nitrogen (g kg^−1)^	1.5 ± 0.4
Total phosphorus (g kg^−1^)	14.3 ± 1.3
Alakayln-hydrolaize N (mg g^−1^)	140 ± 3.3
Organic matter (%)	2.1 + 0.02
Mobile phosphorus (mg kg^−1^)	20 ± 1.2

Mean ± SD.

**Table 2 plants-11-02644-t002:** Soil microbial biomass carbon (MBC) and microbial biomass nitrogen (MBN) of the soil.

Treatmnents	MBC (mg/kg)	MBN (mg/kg)
Control	2.1 ± 0.1 d	40.2 ± 1.1 d
PK	3.6 ± 0.2 cd	49.2 ± 2.11 d
NPK	4.2 ± 1.1 c	107.2 ± 3.07 c
NPK + 40 t manure	6.2 ± 1.3 b	148.4 ± 1.2 b
NPK + 60 t manure	9.4 ± 1.2 a	180.2 ± 5.3 a

Mean ± SD. Lower case letters represent significant difference between treatments at *p* < 0.05.

**Table 3 plants-11-02644-t003:** The effect of mineral fertilizers on the accumulation of biomass in maize in the early period of its development (50 plants/g).

Treatmnents	Biomass (50 Plants/g)
Control	41.8 ± 4.4 d
PK	82.0 ± 2.7 c
NPK	96.0 ± 4.6 b
NPK + 40 m	101.1 ± 2.3 b
NPK + 60 m	112.5 ± 4.8 a

Mean ± SD. Lower case letters represent significant difference between treatments at *p* < 0.05.

**Table 4 plants-11-02644-t004:** Influence of mineral and organic fertilizers on the content of nutrients in the maize plant (% on the dry matter), Arman hybrid.

Stage	Whole Plant			Grain Stage								
	5–7 Leaves Phase			Grain			Leaf Stems			Kernel		
Treatmnents	N	P_2_O_5_	K_2_O	N	P_2_O_5_	K_2_O	N	P_2_O_5_	K_2_O	N	P_2_O_5_	K_2_O
Control	3.34	0.50	4.10	1.44	0.36	0.22	0.53	0.08	1.14	0.57	0.09	0.60
PK	3.53	0.67	4.41	1.57	0.49	0.25	0.62	0.15	1.42	0.52	0.10	0.62
NPK	4.03	0.64	4.41	1.70	0.54	0.23	0.75	0.14	1.20	0.63	0.11	0.71
NPK + 40 tm	4.03	0.68	4.41	1.70	0.51	0.29	0.82	0.13	1.46	0.67	0.12	0.79
NPK + 60 tm	4.00	0.76	4.56	1.80	0.51	0.24	0.82	0.13	1.34	0.68	0.12	0.82

**Table 5 plants-11-02644-t005:** Influence of mineral and organic fertilizers on the yield and protein content.

Treatments	Yield In	Protein Content	Output of Protein
	t/ha	%	t/ha
Control	5.51± 0.19 d	9.0 ± 0.26 d	0.5
PK	6.92 ± 0.21 c	9.8 ± 0.11 c	0.68
NPK	7.87 ± 0.11 b	10.6 ± 1.31 b	0.83
NPK + 40 m	8.47 ± 0.01 a	10.6 ± 0.38 b	0.89
NPK + 60 m	8.49 ± 0.29 a	11.3 ± 0.17 a	0.96

Mean ± SD. Lower case letters represent significant difference between treatments at *p* < 0.05.

**Table 6 plants-11-02644-t006:** Influence of mineral and organic fertilizers on the consumption of nutrients by maize plants.

Phases	Initial (5–7 Leaves)			Full Ripening Phase			Grains		
Treatments	g/50 Plants			kg/ha			kg/ha		
	N	P_2_O_5_	K_2_O	N	P_2_O_5_	K_2_O	N	P_2_O_5_	K_2_O
Control	1.4	0.21	1.71	137	27	144	79	20	12
PK	2.9	0.54	3.61	151	48	175	109	34	17
NPK	3.87	0.61	4.23	236	63	203	133	41	18
NPK + 40 m	4.07	0.69	4.45	278	64	229	144	43	25
NPK + 60 m	4.5	0.86	5.15	294	64	222	153	43	21

**Table 7 plants-11-02644-t007:** Plant height, shoot dry weight, and chlorophyll content of the maize.

Treatments	Plant Height	Dry Weight of Shoot	Chlorophyll
	(cm)	(g/Plant)	mg/g
Control	181 ± 2.3 d	170 ± 5.9 b	4.2 ± 0.1 e
PK	224 ± 10.2 c	189 ± 5.2 a	5.9 ± 0.3 d
NPK	237 ± 5.6 c	199 ± 10.2 a	6.5 ± 0.1 c
NPK + 40 m	262 ± 4.9 b	198 ± 9.1 a	10.2 ± 0.5 b
NPK + 60 m	299 ± 8.5 a	202 ± 7.3 a	12.7 ± 0.2 a

Mean ± SD. Lower case letters represent significant difference between treatments at *p* < 0.05.

## Data Availability

Not applicable.

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
