# Peer review of "Influence of Mineral Fertilizer and Manure Application on the Yield and Quality of Maize in Relation to Intercropping in the Southeast Republic of Kazakhstan"

_plants, 2022, doi:10.3390/plants11192644_

Round 1

Reviewer 1 Report

Major Comments:

Introduction: Add some lines about intercropping, as it is the key point of this issue.

Language: check the language and make corrections if required.

Minor comments

Lines 13: no need for repeated “Kazakhstan”, use once.

Lines 21: Check the format of this line, “t/hat/ha

Lines 26: modify and correct this sentence “With the application of full dose fertilization”

Lines 28: use “significant” sign (p < > 0.05) at the end of the sentence

Lines 35: check the spellings and modify the sentence “caloric intake of almost 4.5 billion people in >”

Lines 47:  modify the sentence, what do you mean by biological “According to its biological”

Lines 55: change this sentence; need correction, “the level of mineral nutrition similarly”

Lines 71: check the line spacing

Lines 72: check the citation format

Lines 87: add the references for “worldwheater”

Lines 87: correct soils to soil

Lines 116: check the words space

Line 121: add the reference for this method

Line 133: check grammar for this sentence

Line 137: add the details for MBC and MBN

Line 142: add ‘g/plants”

Line 1732-173: not clear enough “additional application of N100 on the treatment of P60K100 provided an increase in yield of 0.99”, modify this sentence

Author Response

Dear Reviewer,

                         Please see the attchment.

Reviewer 2 Report

A minimum of 2 years of results is required for this type of field experiment. The interaction of genotype, environmental factors and the evaluated treatments also requires data on weather patterns corresponding to the growing season. 

Regarding methodological inaccuracies, section 2.2 Experimental design and description lacks a description of the experimental design, plot size, harvested area, etc.

Statistical evaluation is poorly described and practically unused.

The numbering of the tables should have been continuous.

Author Response

Dear reviewer,

                         please see the attachement.

Reviewer 3 Report

This manuscript presents an interesting study regarding the response of maize crops to mineral fertilizer and manure application. However, the manuscript requires substantial improvement. Main points

a. In section “Materials and Methods”

Line 95. Table 1 must be modified

Line 97. Experimental design and description:

-Please explain why you have chosen to fertilize with PK only (treatment 2). Do you expect yield improvements compared to the rest of fertilizer doses?

-methods of statistical analysis are not written.

Line 113 Basic characteristics of the soil:

Please include Soil Microbial biomass carbon (MBC) and microbial biomass nitrogen (MBN) of the soil

b. In section “Results”

Significant differences among treatments are not presented in Tables

c. In section “Discussion”

Significant differences among treatments are not presented in Tables

Lines 211-217, 243-245 References are required. Please revise

Lines 263-264 meaning is not clear

Author Response

(The authors gave the same response as above.)

Reviewer 4 Report

The aim of the article might be to clear the effect of mineral fertilizer and manure application on maize production. However, the planning of field examination and the soil sampling is not shown accurately. The information on plot design, such as area and arrangement method, is essential. Methods of soil sampling are also needed. Why didn't you show the condition of the intercrops? The evaluation of the yield data should need more than two growing seasons. 

Author Response

(The authors gave the same response as above.)

Round 2

Reviewer 3 Report

The manuscript has been substantially improved following comments of the reviewers

Author Response

Dear Reviewer,

                        thank you for the valuable suggestions and recommendation, which convert our manuscript in a well presented form.

Kind regards,

Prof. Han Qingfang

Taicheng Road, No. 3, Yangling-712100 Shaanxi, China.

Tel: +86-13609112355

E-mail: hanqf88@nwsuaf.edu.cn

Reviewer 4 Report

Thank you for your response to my comments. I don't have any additional comments.

Author Response

Dear Reviewer,

                        thank you for the feedback, indeed your recommendation were of great significance and were helpful to shape this manuscript in a suitable study.

Kind regards,

Prof. Han Qingfang

Taicheng Road, No. 3, Yangling-712100 Shaanxi, China.

Tel: +86-13609112355

E-mail: hanqf88@nwsuaf.edu.cn